# Relevance of Therapeutic Drug Monitoring of Tyrosine Kinase Inhibitors in Routine Clinical Practice: A Pilot Study

**DOI:** 10.3390/pharmaceutics14061216

**Published:** 2022-06-08

**Authors:** Vanesa Escudero-Ortiz, Vanessa Domínguez-Leñero, Ana Catalán-Latorre, Joseba Rebollo-Liceaga, Manuel Sureda

**Affiliations:** 1Plataforma de Oncología, Hospital Quirónsalud Torrevieja, 03184 Torrevieja, Spain; vanesa.escudero@quironsalud.es (V.E.-O.); ana.catalan@quironsalud.es (A.C.-L.); joseba.rebollo@quironsalud.es (J.R.-L.); 2Pharmacy and Clinical Nutrition Group, Universidad CEU Cardenal Herrera, 03203 Elche, Spain; 3Servicio de Farmacia, Hospital Universitario Morales Meseguer, 30008 Murcia, Spain; vanedole80@hotmail.com

**Keywords:** therapeutic drug monitoring, tyrosine kinase inhibitors, cancer, personalized medicine

## Abstract

Introduction: The main goal of treatment in cancer patients is to achieve the highest therapeutic effectiveness with the least iatrogenic toxicity. Tyrosine kinase inhibitors (TKIs) are anticancer oral agents, usually administered at fixed doses, which present high inter- and intra-individual variability due to their pharmacokinetic characteristics. Therapeutic drug monitoring (TDM) can be used to optimize the use of several types of medication. Objective: We evaluated the use of TDM of TKIs in routine clinical practice through studying the variability in exposure to erlotinib, imatinib, lapatinib, and sorafenib and dose adjustment. Materials and methods: We conducted a retrospective analytical study involving patients who received treatment with TKIs, guided by TDM and with subsequent recommendation of dose adjustment. The quantification of the plasma levels of the different drugs was performed using high-performance liquid chromatography (HPLC). The Clinical Research Ethics Committee of the Hospital Quirónsalud Torrevieja approved this study. Results: The inter-individual variability in the first cycle and in the last monitored cycle was 46.2% and 44.0% for erlotinib, 48.9 and 50.8% for imatinib, 60.7% and 56.0% for lapatinib and 89.7% and 72.5% for sorafenib. Relationships between exposure and baseline characteristics for erlotinib, imatinib, lapatinib and sorafenib were not statistically significant for any of the variables evaluated (weight, height, body surface area (BSA), age and sex). Relationships between height (*p* = 0.021) and BSA (*p* = 0.022) were statistically significant for sorafenib. No significant relationships were observed between C_trough_ and progression-free survival (PFS) or overall survival (OS) for any drug, except in the case of sunitinib (correlation between C_trough_ and PFS *p* = 0.023) in the exposure–efficacy analysis. Conclusions: Erlotinib, imatinib, lapatinib and sorafenib show large inter-individual variability in exposure. TDM entails a significant improvement in exposure and enables more effective and safe use of TKIs in routine clinical practice.

## 1. Introduction

The main goal of pharmacologic treatment in cancer patients is to achieve the highest therapeutic effectiveness with the least iatrogenic toxicity. However, the dosage regimens to achieve this goal can differ considerably from patient to patient. In routine clinical practice, the standardized dosage regimens of the drugs administered are very satisfactory in some patients but may show minimal efficacy or even generate adverse reactions in others [1].

Pharmacokinetic studies the relationship between the dosage regimens used and the corresponding time course of drug and/or metabolite concentrations in the body [2]. The inherent variability in pharmacokinetic processes constitutes one of the main causes of the different clinical responses observed in individual patients.

TDM is the clinical practice of measuring specific drugs at designated intervals to maintain a constant concentration in a patient’s bloodstream, thereby optimizing individual dosage regimens [3]. Drugs typically considered good candidates for TDM were those with a narrow therapeutic window and a clear correlation between exposure and clinical response as antibiotics, antiretrovirals, antiarrhythmics, anticonvulsants, immunosuppressants or antineoplastics.

TKIs are a type of enzyme inhibitor that specifically block the action of one or more tyrosine kinases [4], involving processes such as cell proliferation and survival, transcription, angiogenesis and progression to metastasis. Other effects are mediated by the interaction with therapeutic targets such as epidermal growth factor receptor (EGFR), ATP-binding cassette (ABC) transporters, vascular endothelial growth factor receptor (VEGFR), human epidermal growth factor receptor 2 (HER2), platelet-derived growth factor receptor (PDGFR) or stem cell factor receptor (c-KIT) [5]. TKIs cause fewer non-specific toxicities than standard chemotherapy due to their high affinity for specific molecular mutations of tumor cells [6,7] and can be used both as monotherapy and in combination.

TKIs are usually given at a fixed dose and taken via the oral route. Fixed-dose therapies present a wide range of plasma concentrations, with inter-individual variability in trough concentrations (C_trough_) of up to 23 fold [8]. In the case of subtherapeutic exposure, selection of resistant cell clones can be favored, or the agent inadequately considered non-active. Overexposure can cause undesirable toxicities [9].

Although oral administration implies theoretical advantages to both patients and the health system, evidence suggests that adherence to oral cancer therapies is far from optimal [10,11,12,13,14,15]. Several studies have examined adherence to imatinib treatment in patients with gastrointestinal stromal tumor (GIST) or chronic myeloid leukemia (CML), showing high rates of non-compliance [11,12,16], which can result in a low C_trough_ of imatinib [17]. In patients with CML treated for several years, poor adherence may be the most important cause of not reaching adequate molecular responses [18]. Food–drug interactions and gastrointestinal surgery are other areas of concern. High-fat meals can increase the area under the plasma concentration–time curve (AUC) of nilotinib by up to 80% [19] and that of lapatinib by more than 3 fold [20]. Pazopanib exposure in terms of AUC is doubled when administered with food compared to when administered in the fasted state [21]. Saturation of gastrointestinal absorption has been described for nilotinib at doses greater than 400 mg [22]; and for sorafenib at doses greater than 800 mg [23]. For pazopanib, due to solubility limitations in the digestive tract, a dose higher than 800 mg does not translate into an increase in plasma AUC [24]. Imatinib concentrations were significantly lower in patients undergoing gastrectomy [25,26], possibly due to decreased gastrointestinal transit time or lack of gastric acid secretion [26].

Regarding distribution process, most TKIs are substrates for membrane efflux transporters (e.g., ABCB1 and ABCG2) or uptake transporters (e.g., SLC22A1) [27,28,29,30,31] and present a high affinity for plasma proteins (e.g., 1-acid glycoprotein [AGP] and albumin), with only the free fraction of the drug (corresponding to a small percentage of total plasma levels) exerting any therapeutic action [32,33]. Other emerging parameters to consider are body composition and muscle mass [34,35].

Pharmacogenetics causes part of the high inter-individual variability observed to date in the metabolism of TKIs [8]. TKIs are primarily metabolized by CYP3A4, with a secondary role for other CYP enzymes [27]. Interactions among drugs can explain why the metabolic genotype does not accurately reflect the phenotype and limit the application of pharmacogenomic methods [36]. For all these reasons, dose adjustment is highly recommended for TKIs [37]. Finally, some TKIs have active metabolites for which variations in the enzymes responsible for metabolism translate into variations in plasma concentrations [8].

Most patients with advanced cancer acquire genetically based mechanisms of therapeutic resistance as the disease progresses [38]. Recent advances in high-throughput gene expression profiling enable the identification of differentially expressed genes involved in processes such as drug sensitivity and resistance, signaling pathways relevant to cancer biology, and tumor therapeutic targets in a semi-quantitative and rapid manner. Therefore, it is of interest to explore not only mutations in the genome, but also the expression of multiple genes capable of predicting the outcome of a drug in the clinical setting [39,40].

In general, there is a relationship between the concentration of the drug in plasma and the response and toxicity of the drug. TDM relies on the hypothesis that the concentration of a drug in the blood reflects the concentration at the site of action much better than the administered dose and constitutes a valid tool to complement pharmacogenomic techniques [41,42].

At present, there are more than 30 TKIs available for the treatment of several hematologic and solid tumors. Pharmacokinetic parameters obtained through TDM may be an important biomarker to optimize treatment [43]. Currently, there are numerous ongoing studies of TKIs, usually with highly selected patients. Data on real-world populations are necessary due to widespread use in very different clinical conditions.

The Plataforma de Oncología of Hospital Quirónsalud Torrevieja implemented TDM of TKIs in 2010. Since then, many cases of drug–drug interactions, drug–food interactions, and underdosing or overexposure to the drug at standard doses were detected. The results were analyzed with the aim of evaluating variability in real-world populations and assessing the effectiveness of TDM of TKIs in the routine clinical practice.

## 2. Materials and Methods

### 2.1. Patients

Patients were included in the present study if erlotinib, imatinib, lapatinib or sorafenib was part of their cancer drug therapy in the period 2016–2020. Selection criteria were not previously defined in this study, because the patient cohort should reflect a real-world cohort.

Patients’ medical records were reviewed with the aim of studying all possible factors related to the patient and the medication that may influence the evolution of the plasma concentrations of TKIs. Sociodemographic and anthropometric, pharmacological treatment, clinical, toxicity and therapeutic drug monitoring data were collected. Treatment response was determined according to the RECIST.13. Adverse events were recorded by grade according to the Common Terminology Criteria for Adverse Events version 3.0 (CTCAE v 3.0) [44].

This study was reviewed and approved by the Clinical Research Ethics Committee of the Hospital Quirónsalud Torrevieja and conducted according to Good Clinical Practice guidelines and the Declaration of Helsinki. Patients received information about this study and provided written informed consent.

### 2.2. Treatment and Blood Sampling

Patients included in this study received drugs at doses previously established by their doctors, according to data sheet indications and clinical judgment. In all cases, drug dose was prescribed before the patient enrolled onto this study and was not affected by this study prior to enrollment.

Monitoring was carried out in all cases after at least fifteen days from the start of treatment, when the plasma levels of the drug were considered at steady state. Blood samples were extracted just before ingestion of the tablet/s and at different times after ingestion. The drug was administered in the fasted state and patients were told they could eat food after the second blood sample was drawn. Sampling times, which are defined for each drug in Table 1, were selected based on the optimal sampling theory and pharmacokinetic models previously published in the scientific literature [45,46,47,48]. Blood samples were extracted into tubes with lithium heparin as an anticoagulant and immediately protected from light with aluminum foil. The total blood volume extracted was between 4 and 5 mL for each sample. All samples were centrifuged after extraction at 3500 r.p.m. for 10 min at room temperature and the plasma obtained was frozen at −80 °C until bioanalysis.

Bioanalysis was carried out using high-performance liquid chromatography (HPLC) coupled with ultraviolet (UV) detection. All the techniques used for the quantification of erlotinib, imatinib, lapatinib and sorafenib were previously validated in the Personalized Pharmacotherapy Unit of Hospital Quirónsalud Torrevieja according to the Guidelines of the Food and Drug Administration (FDA) [49] and the European Medicines Agency (EMA) [50], in terms of linearity, precision, accuracy, selectivity, specificity and performance.

#### 2.2.1. Erlotinib

As the stationary phase, an Ultrabase^®^ C_18_ chromatographic column, 4.6 mm in diameter by 150 mm in length, made of stainless steel and filled with a C_18_ reversed-phase siliceous substrate with a particle size of 5 µm, was used. A mixture of 40% of 0.02 M ammonium acetate, pH = 7, and 60% acetonitrile: methanol at a ratio of 70:30 (*v*/*v*) was used as the mobile phase. The linearity of the technique included a concentration range of 0.05–7.5 µg/mL. The limit of quantification was 0.5 µg/mL. Solid–liquid extraction was used and the analysis time was 9 min.

#### 2.2.2. Imatinib

The stationary phase used was an Ultrabase^®^ C_18_ chromatographic column, 4.6 mm in diameter by 150 mm in length, made of stainless steel and filled with a C_18_ reversed-phase siliceous substrate with a particle size of 5 µm. The mobile phase used was a 40% mixture of 0.02 M ammonium acetate, pH = 7, and 60% acetonitrile: methanol at a ratio of 70:30 (*v*/*v*). The linearity of the technique included a concentration range of 0.05–5 µg/mL. The limit of quantification was 0.5 µg/mL. Solid–liquid extraction was used and the analysis time was 10 min.

#### 2.2.3. Lapatinib

The stationary phase used was a Kromasil^®^ C_18_ chromatographic column, 4.6 mm in diameter by 150 mm in length, made of stainless steel and filled with a C_18_ reversed-phase siliceous substrate with a particle size of 5 µm. The mobile phase used was a mixture of acetonitrile and 0.02 M ammonium acetate, pH = 3.5, at a ratio of 53:47 (*v*/*v*). The linearity of the technique included a concentration range of 0.02–10 µg/mL. The limit of quantification was 0.2 µg/mL. Liquid–liquid extraction was used and the analysis time was 12 min [51].

#### 2.2.4. Sorafenib

The stationary phase used was a Kromasil^®^ C_18_ chromatographic column, 4.6 mm in diameter by 150 mm in length, made of stainless steel and filled with a C_18_ reversed-phase siliceous substrate with a particle size of 5 µm. The mobile phase used was a mixture of acetonitrile and 0.02 M ammonium acetate, pH = 3.5, at a ratio of 53:47 (*v*/*v*). The linearity of the technique included a concentration range of 0.1–20 µg/mL. The limit of quantification was 0.1 µg/mL. Liquid–liquid extraction was used and the analysis time was 12 min [52].

### 2.3. Pharmacokinetic Analysis

#### 2.3.1. Erlotinib

The pharmacokinetic analysis of erlotinib, like that of imatinib, lapatinib and sorafenib, was performed with the NONMEM VII version 2.0 software (ICON, Hanover, MD, USA) using the POSTHOC option [53]. The program was compiled with the DIGITAL Visual Fortran version 6.6C program. The graphics were made with the S-Plus 6.1 Professional Edition program for Windows (Insightful, Seattle, WA, USA). A one-compartment pharmacokinetic model with first-order absorption and elimination kinetics was selected to describe erlotinib plasma concentrations, as performed previously by other authors [45]. The fixed-effect parameters estimated by the model were V (volume of distribution, L), CL (plasma clearance, L/h) and K_a_ (absorption constant). The model included plasma ALT concentration and age as covariates in CL, and weight in V.

#### 2.3.2. Imatinib

To describe the plasma concentrations of imatinib, a one-compartment pharmacokinetic model with zero-order absorption and first-order elimination kinetics was selected, as performed previously by other authors [46]. The model was parameterized in terms of D1 (zero-order absorption duration), V and CL, and included the plasma concentration of α 1-acid glycoprotein as a covariate in CL.

#### 2.3.3. Lapatinib

In the case of lapatinib, a one-compartment pharmacokinetic model with first-order absorption and elimination kinetics was selected, as performed previously by other authors [47]. The fixed-effect parameters estimated by the model were V, CL and K_a_. The model included plasma ALT concentration and age as covariates in CL, and weight in V.

#### 2.3.4. Sorafenib

Sorafenib plasma concentrations were described using a one-compartment pharmacokinetic model previously used by Jain L et al. to describe the pharmacokinetics of sorafenib [48]. The model uses 4 absorption transit compartments, enterohepatic circulation, and first-order elimination kinetics.

### 2.4. Statistical Methods

The statistical analysis was carried out with the SPSS software (version 20.0 for Windows^®^, Chicago, IL, USA). All data were stored in the database and filtered through distributions of unknown values for every variable and through distributions to detect uncommon values. Two-sided significance tests were used in the analyses performed, considering a probability of error α (*p* < 0.05) as significant.

For the descriptive analysis of the data, the categorical variables were expressed as a percentage, while the continuous variables were expressed as the mean (standard deviation (SD)). To describe the normality in the distribution of continuous variables, the distribution profile of the values of each variable was evaluated through histograms and trend lines, in addition to performing the Kolmogorov–Smirnov test. To validate the assumption of homogeneity of variances, Levene’s test was used.

A linear regression model with an established Pearson’s correlation coefficient (r) between continuous variables was used for the analytical treatment of the data.

To compare the mean values of continuous variables in different events, Student’s *t*-test was used for those variables with a normal distribution and the Mann–Whitney U test was used for those that did not follow a normal distribution.

To compare the variation in categorical variables (rash, diarrhea, fatigue, abdominal pain, hypertension, and mucositis) between the first and the last monitored cycle, the non-parametric chi-square test was performed.

To assess the relationship between treatments protocols and disease progression or mortality, a Kaplan–Meier survival curve was plotted.

## 3. Results

### 3.1. Patients

The plasma levels of erlotinib, lapatinib, imatinib or sorafenib were monitored in 58 patients (57% women and 43% men) receiving 141 cycles of each drug (mean 2.4 cycles/patient, range 1–12) for dose individualization. A summary of all baseline characteristics of patients is shown in Table 2.

A total of 710 plasma samples (307 for erlotinib, 137 for lapatinib, 114 for imatinib and 152 for sorafenib) were analyzed according to sampling times detailed in Table 1. We observed large interpatient variability for all drugs, as shown in Figure 1.

The mean interpatient variability in dose-normalized plasma concentrations in the first monitored cycle, expressed as a coefficient of variation (CV, %), was 46.2%, 60.7%, 48.9% and 89.7%, for erlotinib, lapatinib, imatinib and sorafenib, respectively. After TDM and dose adjustments, the mean interpatient variability in dose-normalized plasma concentrations in the last monitored cycle was 44%, 56%, 50.8% and 75.5%, respectively.

No significant effect of body weight, age, height or BSA on TKI plasma concentrations was observed, except for a negative correlation between height (r = −0.4, *p* = 0.021) and BSA (r = −0.4, *p* = 0.022) versus plasma concentrations of sorafenib. A non-significant increase in the C_trough_ of drugs was observed in women vs. men.

#### 3.1.1. Erlotinib

In 21 of the monitored cycles (38.2%), the recommendation was to increase the prescribed dose. A total of 5 of the 22 patients included in this study required a mean dose increase of 46.6% in the last monitored cycle compared to the first one. In 30 of the monitored cycles (54.5%), the recommendation was to maintain the current dose since the concentrations were within the therapeutic target interval. In two of the cycles (3.6%), it was recommended to decrease the dose, with a dose reduction of 24.9% in the last monitored cycle with respect to the first one in 2 patients. In addition, in two of the monitored cycles (3.6%), it was recommended to suspend administration of the drug.

#### 3.1.2. Imatinib

In seven of the monitored cycles (31.8%), the recommendation was to increase the prescribed dose since the concentrations were below the target therapeutic range—this meant a mean dose increase of 77.7% in the last monitored cycle compared to the dose administered in the first one for 3 of the 9 patients included. In 15 of the monitored cycles (68.2%), the recommendation was to maintain the current dose since the concentrations were within the therapeutic range. There were no recommendations to reduce the dose of imatinib, since no concentrations were found above the target level.

#### 3.1.3. Lapatinib

In five of the monitored cycles (14.3%), the recommendation was to increase the prescribed dose—this meant a mean increase of 175% in the last monitored cycle with respect to the first one for 2 of the 16 patients included. In 24 of the monitored cycles (68.6%), the recommendation was to maintain the current dose since the concentrations were within the therapeutic range. In three of the cycles (8.6%), it was recommended to reduce the initial dose for concentrations above the therapeutic range, with a dose reduction of 62.5% of the dose administered in the last cycle with respect to the first one in 2 of the patients included. In three of the monitored cycles (8.6%), the recommendation was to suppress the drug.

#### 3.1.4. Sorafenib

In five of the monitored cycles (17.2%), the recommendation was to increase the prescribed dose since the concentrations were below the therapeutic range, with a mean increase of 200% in the last compared to the first one in 2 of the 11 patients included. In 21 of the monitored cycles (72.4%), the recommendation was to maintain the current dose. In two of the cycles (6.9%), it was recommended to decrease the initial dose since concentrations were above the therapeutic range, with a reduction of 50% of the dose in the last one with respect to that in the first one in 1 patient. In one cycle (3.4%), it was recommended to suspend administration of the drug.

### 3.2. Response and Survival

Table 3 shows the survival data in terms of PFS and OS. The maximum follow-up period was 100 months for erlotinib, 176 months for imatinib, 105 months for lapatinib, and 62 months for sorafenib.

No significant relationship between PFS and C_trough_ (*p* > 0.203 in all cases) or between OS and C_trough_ (*p* > 0.251 in all cases) was observed for none of the TKIs under study.

### 3.3. The Exposure–Toxicity Relationship

A total of 33 of the patients included in the present study developed toxicity at the start of treatment: 9 erlotinib patients (45.4%), 5 imatinib patients (55.5%), 12 lapatinib patients (75%), and 7 sorafenib patients (63.3%). The toxicity incidence data for each drug are detailed in Table 4.

Patients were classified into four groups to assess the toxicities developed based on exposure to each of the drugs: (1) patients with drug concentrations below the target level and without toxicities, (2) patients with drug concentrations below the target level and with toxicities, (3) patients with drug concentrations above the target level and without toxicities and (4) patients with drug concentrations above the target level and with toxicities. The distribution of patients according to the four groups described above is shown in Figure 2.

## 4. Discussion

TDM aims to maximize the therapeutic effectiveness of pharmacologic treatments while reducing the iatrogenic toxicity produced by them. TDM was typically considered advantageous for drugs with a large inter-individual variability in exposure with relatively low intra-individual variation, a significant exposure–efficacy relationship, a narrow therapeutic window, and availability of a validated bioanalytical assay [22]. It has been postulated recently that this could also represent a useful tool to individualize dosing and optimize treatment using drugs with a wide therapeutic window and high cost [54,55,56].

TKIs are drugs with a narrow therapeutic window and high cost which are commonly administered at fixed doses. Even though TKIs present high variability in their pharmacokinetics, which translates into high inter- and intra-individual variability in drug exposure as stated above [57,58], the standard dose described in the corresponding data sheet of the drugs is administered to all candidate patients, resulting in unpredictable plasma concentrations [27]. Clinical trials do not identify all real-world scenarios because frail patients, those on specific diets, variable dosage intervals, or concomitant polimedication are systematically excluded from them. Post-marketing studies detect or better describe different side effects that can decrease adherence to the treatment, reducing the efficacy of these drugs. TDM can improve tolerability and subsequently adherence [59].

Oral cancer therapies have theoretical advantages, both for patients and the healthcare system, but not without drawbacks. Adverse effects increase therapeutic non-compliance and symptoms of anxiety and depression. These, in turn, cause loss of control of the disease, higher costs due to increased and longer admissions in the hospital, and loss of quality of life. TDM, in this context, enables identification inadequate dosing, adapting the dose to the individual needs of the patient and avoiding abandonment of therapy due to toxicity or futility and unjustified changes in therapy caused by low plasma levels of the drug [59].

The present evaluation has been performed on adult real-world patients, with no inclusion or exclusion criteria other than an expected efficacy of the TKI in their clinical situation. They presented high variability in terms of tumor type, disease stage, pre-treatment, duration of treatment, etc. In this population, the variability in drug exposure can be much greater than in clinical trials and the response to treatment, both in terms of efficacy and toxicity, reflects it. This enabled the detection of personalized influences of different factors on exposure, response, and toxicity of TKI treatments and highlighted the benefit of TDM in routine clinical practice.

### 4.1. Erlotinib

In the case of erlotinib, data on underdosed patients found in the present study differed from those obtained by Lankheet et al. [60], who used a plasma concentration of 500 mg/L as the therapeutic target [61] and found that 11.1% of patients had plasma concentrations below the target, a percentage of underdosed patients that is clearly lower than that reported here. In this study, all the patients included had a diagnosis of lung cancer, so this lower percentage of underdosed patients may be due to lower interpatient variability.

The data sheet of erlotinib [62] recommend a dose of 150 mg/day for adult patients with non-small-cell lung cancer (NSCLC) and 100 mg/day for those with pancreatic cancer. In the present study, patients started with standard doses, but five of them received higher doses in the last cycle because they could not otherwise reach therapeutic levels within the optimal range. Erlotinib dose adjustment is only recommended in the data sheet when co-administered with CYP3A4 inhibitors or inducers, CYP1A2 inhibitors, or in smokers, but, as shown, dose adjustments based on TDM are necessary in other situations.

No effect of weight, height, BSA, age or sex on the plasma C_trough_ was detected in the present study, as was the case in the patients studied by Lankheet et al. [60] and other population studies involving 1859 NSCLC patients [63], 291 NSCLC patients [62], 80 NSCLC patients [64] and 204 pancreatic cancer patients [62].

A direct relationship between exposure to erlotinib and observed clinical response must be proven [65]. Some authors describe a statistically significant correlation between exposure and efficacy [66,67], while others only point to a non-statistically significant trend or relationship between both parameters [68,69]. The preclinical study carried out by Hidalgo et al. [61] suggested that a target C_trough_ greater than 500 mg/L in humans would be adequate for the inhibition of EGFR receptors, so this target concentration was adopted for later studies.

In the present study, PFS is similar (8 months) to previously published results, whereas OS is greater (32 months). Motoshima et al. found PFS and OS of 6.3 months and 16.9 months, respectively, in 26 NSCLC patients [67]. The differences observed may be due to the smaller number of patients recruited in this study, their complexity and underlying pathogenesis, and the follow-up time.

No statistically significant relationship was found between dose-normalized C_trough_ and PFS (*p* = 0.630) or OS (*p* = 0.593). Similarly, Tiseo et al., in their study of 56 patients with NSCLC, found no significant relationship between a C_trough_ ≥ 4.6 nmol/mL or greater skin toxicity (patients with greater skin toxicity had better results from treatment) with an improvement in OS (*p* = 0.351) or PFS (*p* = 0.127) [66]. In a phase II study involving 16 NSCLC patients, C_trough_ was measured on days 2 and 8 of treatment. The ratio of C_trough_ on day 8 to C_trough_ on day 2 represented the accumulation of erlotinib over time. A high ratio was related to a low metabolism of erlotinib and therefore higher exposure, and correlated positively with PFS (*p* = 0.004) but not with OS [67]. Another study of patients with HNSC evaluated three ranges of samples depending on the time elapsed after taking the drug—in 42 patients, the C_trough_ was between 20 and 25 h after the dose; in 77 patients, C_max_ was between 2 and 5 h post dose; in 47 patients, the concentration was stated for between 5 and 10 h post dose. The median concentration between 5 and 10 h post dose for erlotinib predicted better OS (*p* = 0.021) [68].

The main analytical parameters related to toxicity (hemoglobin, hematocrit, neutrophils, leukocytes, platelets, creatinine, bilirubin, GOT, GPT and alkaline phosphatase), showed no significant differences in the first monitored cycle with respect to the last, as with the defined toxicities themselves (fatigue, diarrhea, abdominal pain, hypertension, mucositis, anemia, neutropenia, thrombocytopenia, and renal insufficiency (RI)). Most of the grade 3 toxicities described were in the first monitored cycle (rash, mucositis and thrombocytopenia), before carrying out TDM and dose adjustment, compared to the last monitored cycle (13, 6% vs. 7.1%, 4.5% vs. 0%, and 4.5% vs. 0%, respectively). Grade 2 skin rash was the only relevant toxicity observed in a higher percentage of patients in the last monitored cycle (13.6% in the first monitored cycle vs. 14.3% in the last). This may be due to the relationship between the skin rash produced by erlotinib and tumor response, for which the rash is a potential marker of activity [68,69,70,71]. Development of cutaneous rash has been related to drug exposure and clinical benefit with erlotinib. Soulieres et al. [68] reported higher OS in patients who developed a grade ≥2 skin rash (*p* = 0.045, *n* = 115). In addition, the C_trough_ in this group of patients was higher, although not statistically significant, than the C_trough_ in patients with a rash grade lower than 2 (1097–1126 mg/L vs. 803 mg/L, *p* = 0.49).

No significant relationship was found between the different degrees of skin rash and C_trough_ (rash G1 *p* = 0.870, rash G2 *p* = 0.746 and rash G3 *p* = 0.374) in the present study. In a phase II study of 57 NSCLC patients, the OS of patients with grade 2 rash was 19.6 months, the OS of patients grade 1 rash was 8.5 months and the OS of patients without rash was 1.5 months [72]. Other groups reported similar results [66,67,68,69]. Despite some studies showing a relationship between pharmacokinetic parameters and efficacy, and between toxicity and treatment efficacy, pharmacokinetic parameters did not correlate with toxicity in all cases [66,67,68]. This indicates that the skin rash is not simply a reflection of exposure to erlotinib, in accordance with the results in this study. The largest study to determine this relationship was carried out in 339 NSCLC patients, demonstrating a relationship between AUC_0–24_ and C_max_ with the appearance of rash. However, the correlation was not considered relevant, as there was a large overlap in AUC_0–24_ and C_max_ between patients with and without toxicity [63].

The only significant correlation between the analytical parameters collected and C_trough_ was an inversely proportional relationship with alkaline phosphatase (*p* = 0.025). There are studies that show a relationship between the pharmacokinetic parameters of erlotinib and liver function, although none of them relates it with plasma concentrations [63]. These results led to the recommendation of dose adjustment of erlotinib in hepatic insufficiency.

The relationship between C_trough_ and grade 1 RI was the only statistically significant (*p* > 0.013) relationship between C_trough_ and a toxicity (fatigue, diarrhea, abdominal pain, hypertension, mucositis, anemia and thrombocytopenia). It was not considered clinically relevant, since the 10 patients who presented with grade 1 RI already presented with this at the beginning of this study, in the first monitored cycle. In the literature, we have not found studies showing a relationship between RI and C_trough_, although it seems logical to think that, since only 9% of a single dose is eliminated in the urine [73], there is no relationship between renal function and the pharmacokinetic parameters of erlotinib.

### 4.2. Imatinib

Imatinib showed large variability both in the first and in the last monitored cycle (mean CV first cycle 48.91% vs. mean CV last cycle 50.84%). In different populations, the pharmacokinetic analysis revealed that weight, age, sex, diagnosis, AGP, albumin, granulocytes, white series, hemoglobin and gastrectomy might be factors that explain the variability between patients, but they have not been considered significant for dose adjustments [26,28,33]. As with other TKIs, this variation can be explained by the oral route, whether the drug is administered with food, presystemic metabolism at the intestinal level, membrane transporters, binding to plasma proteins or hepatic metabolism through cytochrome CYP3A4 and CYP3A5. In addition, therapeutic compliance can vary between patients or even in the same patient [2,27]. Patients included in the present study show variability in drug exposure comparable to that observed in other studies [60].

All imatinib dose adjustments made following TDM indications were to increase the dose. In 31.8% of cycles, plasma concentrations were below the established 50% population prediction interval. Dose adjustment in these patients led to a mean increase of 77.7% in the last monitored cycle compared to the first. There are studies confirming the fact that after 8 months of treatment with imatinib, exposure to the drug decreases due to the induction of liver enzymes [60,74].

As detailed in the data sheet of imatinib [75], the recommended dose for adult patients ranges from 100 to 800 mg/day, depending on the indication. Although the patients included started with the standard dose, three of them had to receive higher doses because they did not reach therapeutic levels with the initial doses. Dose adjustment is only recommended in the data sheet when co-administered with inhibitors or inducers of CYP3A4, with liver dysfunction or RI [75]. As evidenced in this study, dose adjustments based on TDM are necessary in additional situations where optimal therapeutic levels are not reached if the patient’s condition allows it.

No significant relationship was observed between dose-normalized plasma C_trough_ in every cycle and PFS and OS. Other authors found that patients with lower exposure to imatinib showed a lower overall rate of benefit (complete response + partial response + stable disease), suggesting that a determinate level of C_trough_ is necessary to maintain the response in patients with GIST [76]. Another prospective study demonstrated a decrease of approximately 30% in C_trough_ after 3 months of treatment [74]. Therefore, it would be advisable to repeat TDM after 3 months of treatment. Widmer et al. demonstrated the relevance of achieving adequate levels of imatinib to maintain therapeutic responses in a study of 38 patients with GIST [33]. This work also suggested that sustained exposure is associated with a better response, more than total exposure to the drug. In the present study, no relationship was found between C_trough_ and PFS or OS, probably due to the low number of patients recruited.

No significant differences were found in the analytical parameters or defined toxicities in the first monitored cycle with respect to the last. Grade 2 toxicities observed, specifically grade 2 anemia, were in the first cycle, before TDM and dose adjustment, compared to the last (22.2% vs. 16.7%), except grade 2 RI (11.1% vs. 33.3%). Chronic RI is described in the data sheet as an adverse reaction of unknown frequency for imatinib [75]. There was no significant correlation between the analytical parameters and C_trough_, except for hemoglobin and hematocrit, where a significant and inversely proportional relationship with C_trough_ was observed. Other authors reported a direct relationship between C_trough_ and analytical alterations or some toxic manifestations, probably related to longer treatment or higher casuistry [77,78].

### 4.3. Lapatinib

Patients treated with lapatinib in this study had different tumors: breast cancer, GIST, pancreas, colon, esophagus, etc. Due to this heterogeneity, the dosage in the monitored treatment cycles varied between 125 mg/day in some patients and 1250 mg/day. There is no similar study in the literature on a real-world population for comparison.

Large variability was observed in the dose-normalized plasma concentrations at different sampling times, both in the first cycle and in the last (mean CV first cycle: 60.7% vs. average CV last cycle: 56.0%), in concordance with other TKIs previously analyzed. Interpatient variability in lapatinib exposure found in the literature shows data for C_trough_ and an AUC of 55–97% and 42–117%, respectively [20,79,80,81,82], consistent with that obtained in the present study. Regarding variability in intra-patient exposure, EPAR data for lapatinib show a CV of AUC between 30% and 36% on average in healthy subjects. There are no other studies in which variability has been measured. Therefore, patients included in this study show interpatient variability in drug exposure comparable to the few studies available in the literature. As other authors have shown for other TKIs, this variability may be influenced by patients’ adherence to treatment, concomitant medication or previous lines of treatment received, etc. [60].

As detailed in the data sheet of lapatinib [79], the recommended dose in adult patients with HER2-positive breast cancer is 750 to 1500 mg/day, depending on the concomitant medication administered. Dose adjustment is only recommended when cardiac events, diarrhea or other grade 2 toxicities occur. In the case of severe hepatic insufficiency, it is recommended to suspend treatment and not restart it. In the present study, 7 of the 16 patients treated began at a reduced dose for impaired liver function. Only in 2 of the patients who started at a reduced dose was it necessary to increase the dose for inadequate plasma levels. In the remaining 5 patients with a reduced dose, optimal levels within the 50% population prediction interval were observed. In patients with impaired liver function, TDM enabled the quantification of drug exposure and subsequent satisfactory dose adjustment.

In situations other than impaired liver function in which optimal therapeutic levels are not reached or when they are reached with doses lower than the standard, as long as the patient’s condition allows it, dose adjustments based on TDM are necessary. Even so, based on the data available in the literature, the individualization of the dose of lapatinib with TDM based solely on a target dose is not recommended because they are studies carried out in a very limited and heterogeneous population [57,58].

In the present study, no significant relationship was observed between dose-normalized plasma C_trough_ at each monitored cycle and PFS or OS. A clear exposure–efficacy relationship for lapatinib has not been identified in the literature. Lapatinib was found to be well tolerated at doses ranging from 175 to 1800 mg once daily or 500 to 900 mg twice daily [81]. In a phase I trial, it was found that most patients who responded to lapatinib had a C_trough_ ranging from 300 to 600 mg/L (*n* = 67 patients with metastatic solid tumors) [81]. However, the results are difficult to interpret as response data are limited and the population is highly heterogeneous.

The analytical parameters related to toxicity in the first monitored cycle with respect to the last showed no significant differences, as with the defined toxicities themselves. The most serious side effects observed, at grade 2 and grade 3, were mostly in the first cycle (rash, diarrhea, mucositis, anemia and neutropenia), before performing TDM and dose adjustment, compared to the last monitored cycle. It is remarkable in the case of lapatinib that toxicities were detected in 25 cases in the first monitored cycle, while toxicities were only detected in 5 cases after performing TDM. TDM of lapatinib reduced toxicity events presented by the study population by 80%, very important in an otherwise useful drug with a high degree of therapeutic withdrawal due to toxicity.

GOT and GPT were the only analytical parameters that correlated with C_trough_, showing a directly proportional relationship. These data are consistent with those found in the literature. Higher systemic exposure to lapatinib was previously described in patients with severe liver failure, with a mean AUC of lapatinib increased by more than 60% and a 3-fold higher t_1/2_ than that observed in patients with normal liver function [79,80]. No relationship was found between fatigue, rash, diarrhea, abdominal pain, hypertension, mucositis or thrombocytopenia and C_trough_ (*p* > 0.093). Other studies described similar results, with diarrhea directly and positively related to dose (*p* < 0.03) but not to C_trough_. This fact suggests that diarrhea is caused by a local effect of the drug on the intestinal epithelium [81,82].

### 4.4. Sorafenib

Due to the different origin of the tumors treated with sorafenib (liver, breast, oral cavity, etc.), the dosage guidelines varied from 200 mg/day in some patients to 600 mg/8 h in other cases. There is no study in the literature that includes a similar real-world population with TDM data for comparison with that shown here.

The large variability observed in dose-normalized plasma concentrations at different sampling times in previous TKIs analyzed was also observed in the case of sorafenib (mean CV first cycle: 89.7% vs. mean CV last cycle: 72.5%). In addition to other already mentioned factors, sorafenib intake is recommended outside of meals or with a moderate or low-fat meal due to the influence of food in pharmacokinetics. It also presents pre-systemic metabolism at the intestinal level. The variability in exposure between patients found in the literature shows data for C_trough_, AUC and apparent oral Cl of 25–104%, 12–117% and 13–80%, respectively [83,84,85,86]. Therefore, the variability in drug exposure in patients included in this study is comparable to that observed in other previously published studies.

The data sheet of sorafenib recommend a dose of 400 mg twice daily for adult patients with hepatocellular carcinoma, renal cell carcinoma or differentiated thyroid carcinoma [83]. In the present study, only two patients started at the standard doses, while nine of the patients started at a reduced dose, either because it was administered in combination with chemotherapy, due to their age (78-year-old patient), as a result of cumulative toxicity, or because they were in a bad clinical condition. One of the patients who started at a reduced dose required an increase from 400 mg/day to higher than 1800 mg/day (350%) to reach levels within the therapeutic range. Another patient required a 50% reduction from 400 mg/day due to levels above the population prediction interval. (Sorafenib dose adjustment is recommended in the data sheet only to manage possible toxicities or with co-administration of neomycin or other antibiotics that cause ecological alterations in the gastrointestinal microflora and may lead to decreased bioavailability or if the drug is administered with inducers of metabolic enzymes. As has been observed in this study, it is necessary to make dose adjustments based on TDM in situations other than suggested if the patient’s condition allows it.

The correlation analysis of height and BSA with plasma C_trough_ shows a positive relationship in both cases (*p* = 0.021 and *p* = 0.022, respectively). No studies assessing the clinical significance of this relationship were found.

In the present study, no significant relationship was observed between dose-normalized plasma C_trough_ at every monitored cycle and PFS (*p* = 0.940) or OS (*p* = 0.909). A clear relationship between exposure and efficacy has not been established at present for sorafenib, with some studies showing a relationship [87,88] and others not doing so [27].

The difference between the first and last cycles monitored was significant only for hemoglobin (*p* = 0.031) and hematocrit (*p* = 0.031) when hematological parameters were analyzed according to exposure to sorafenib. The most remarkable toxicities observed were anemia and grade 3 neutropenia, which mainly occurred in the last cycle (50% and 25%, respectively), that is, after performing TDM and dose adjustment. Hematological toxicities such as anemia or neutropenia are described as frequent in the data sheet of sorafenib after continued administration of the drug, even at levels within the therapeutic range [83]. The correlation study performed between C_trough_ and the different analytical parameters or toxic manifestations show statistically significant results for platelets (r = −0.483, *p* = 0.043), GPT (r = −0.478, *p* = 0.045) and alkaline phosphatase (r = −0.746, *p* = 0.001). Most of the studies which relate exposure and toxicity to sorafenib use AUC as a pharmacokinetic parameter [34,87,89], so they are not comparable to the present study. Fukudo et al. showed that C_trough_ at steady state correlates with grade 2 hand–foot syndrome (*p* = 0.0045) and hypertension (*p* = 0.0453). Although the toxicities they described were not the same as those related in the present work, an exposure–toxicity relationship was shown in both studies [88].

In the study by subgroups of the toxicities for each TKI evaluated, patients in group 1 (patients with concentrations below therapeutic levels and no toxicities) and group 4 (patients with levels above the therapeutic range and toxicities), the results are similar to those reported in the literature for erlotinib and imatinib [60]. The authors conclude that 65.7% of the patients treated with erlotinib and 47.3% of those treated with imatinib would benefit most from TDM as a tool to improve the exposure–efficacy and exposure–toxicity relationships. These percentages were slightly different from ours (63.7% and 55.5%, respectively), but significant enough to consider performing TDM. A total of 75.9% and 36.4% of lapatinib and sorafenib patients, respectively, would benefit most from TDM. There is no similar study in the literature to compare these results.

## 5. Conclusions

Erlotinib, imatinib, lapatinib and sorafenib show high inter-individual and intra-individual variability in exposure. A similar scenario might be anticipated for other TKIs.

TDM of TKIs adds remarkable value to routine clinical practice. TDM enables assessment of underdosing, lack of adherence to treatment and changes in exposure due to interaction with food or other drugs. TDM-guided dose corrections result in a significant improvement in exposure and enable more effective and safe use of TKIs, avoiding the ineffectiveness and toxicity of these drugs in certain patients and clinical situations, as has been the current practice with other medications (antibiotics, digoxin, anticonvulsants, etc.).

## Figures and Tables

**Figure 1 pharmaceutics-14-01216-f001:**
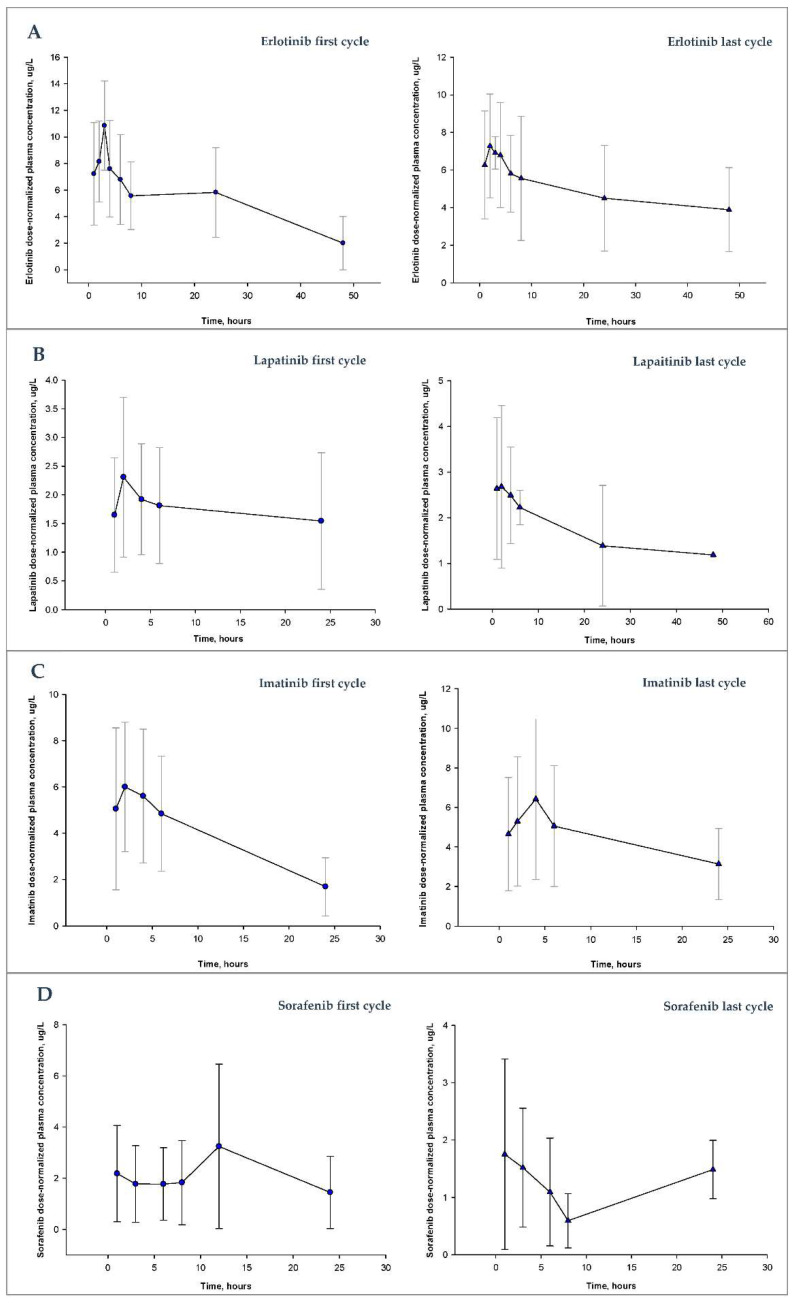
Dose-normalized plasma concentration profiles after administration of drug to patients receiving the dose of erlotinib (**A**), lapatinib (**B**), imatinib (**C**) or sorafenib (**D**) in the first or last monitored cycle. The symbols (circles and triangles) represent the mean dose-normalized plasma concentrations of patients at different times. The error bar for each point represents the standard deviation for each mean.

**Figure 2 pharmaceutics-14-01216-f002:**
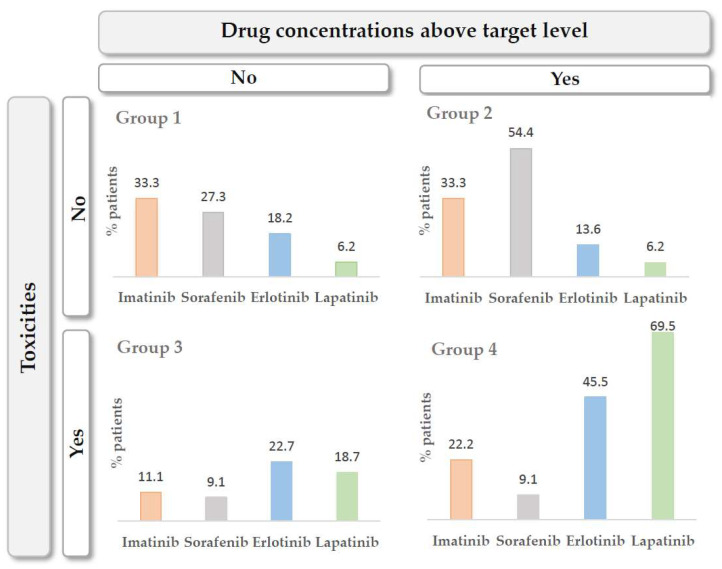
Groups of patients according to exposure to the drugs under study.

**Table 1 pharmaceutics-14-01216-t001:** Blood sampling times for monitored TKIs.

Drug	Sampling Times
Erlotinib Imatinib Lapatinib	Before drug administration and then at 1, 2, 4 and 6 h after drug administration
Sorafenib	Before drug administration and then at 1, 3, 6 and 8 h after drug administration

**Table 2 pharmaceutics-14-01216-t002:** Baseline characteristics.

Baseline Characteristics	Treatment			
	Erlotinib	Lapatinib	Imatinib	Sorafenib
Number of Patients in Treatment	22	16	9	11
Number of total cycles monitored	55	35	22	29
Cycles monitored by patient (n, %)				
1 cycle	8 (38.1)	9 (56.3)	3 (33.3)	7 (63.6)
2 cycles	6 (28.6)	4 (25.0)	2 (22.2)	1 (9.1)
3 cycles	3 (13.6)	2 (12.5)	2 (22.2)	1 (9.1)
4 cycles	2 (9.5)	--	1 (11.1)	1 (9.1)
>5 cycles	3 (14.3)	1 (6.3)	1 (11.1)	1 (9.1)
Gender (n. %)				
Male	14 (63.6)	4 (25.0)	3 (33.3)	4 (36.4)
Female	8 (36.4)	12 (75.0)	6 (66.7)	7 (63.6)
Age (mean (SD). years)	63.0 (12.3)	54.5 (12.6)	50.8 (16.4)	57.1 (15.6)
Weight (mean (SD). Kg)	80.2 (14.6)	69.7 (11.0)	78.50 (18.2)	71.2 (13.3)
Size (mean (SD). cm)	169.7 (9.3)	163.8 (8.4)	168 (11.5)	166.6 (12.2)
Body surface area (mean (SD). m^2^)	1.9 (0.2)	1.7 (0.1)	1.87 (0.3)	1.8 (0.2)
Metastasis (number of patients. %)	19 (86.4)	15 (93.8)	5 (55.6)	9 (81.8)
Previous lines of treatment (range)	1.64 (0–7)	2.4 (1–6)	0.56 (0–2)	1.91 (0–5)

n: number of patients. SD: standard deviation.

**Table 3 pharmaceutics-14-01216-t003:** Survival analysis.

Drug	Progression-Free Survival (Months)			Overall Survival (Months)		
	Median	SE	CI 95%	Median	SE	CI 95%
Erlotinib	8	4.7	0.0–17.1	32	31.6	0.0–93.9
Imatinib	28	11.9	4.6–54.4	90	29.9	31.3–148.0
Lapatinib	8	3.0	2.1–13.9	46	18.9	9.0–83.0
Sorafenib	9	3.4	2.3–15.6	9	5.1	0.1–19.1

SE: Standard error. CI 95%: 95% confidence interval.

**Table 4 pharmaceutics-14-01216-t004:** Incidence of toxicity.

Characteristic	Erlotinib		Imatinib		Lapatinib		Sorafenib	
Patients with toxicity [n. (%)]		9 (45.4)		5 (55.5)		12 (75.0)		7 (63.3)
Toxicity [n. (%)]	RI G1	9 (45.4)	Anemia G2	2 (22.2)	RI G1	6 (37.5)	Abdominal pain	2 (18.2)
	Skin rash G1	5 (22.7)	RI G1	2 (22.2)	Anemia G1	3 (18.7)	IR G1	2 (18.2)
	Anemia G1	4 (18.1)	Fatigue G1	1 (11.1)	Diarrhea G1	2 (12.5)	Anemia G1	2 (18.2)
	RI G2	4 (18.1)	Anemia G1	1 (11.1)	Pain	2 (12.5)	Diarrea G2	1 (9.1)
	Skin rash G2	3 (13.6)	RI G2	1 (11.1)	Anemia G2	2 (12.5)	Anemia G2	1 (9.1)
	Skin rash G3	3 (13.6)			Neutr. G3	2 (12.5)	Thromb. G1	1 (9.1)
	Diarrhea G2	3 (13.6)			RI G2	2 (12.5)		
	Fatigue G2	2 (9.1)			Skin rash G1	1 (6.2)		
	Anemia G2	2 (9.1)			Skin rash G3	1 (6.2)		
	Mucositis G3	1 (4.5)			Diarrhea G2	1 (6.2)		
	Thromb. G3	1 (4.5)			Diarrhea G3	1 (6.2)		
					Mucositis G3	1 (6.2)		
					Anemia G3	1 (6.2)		

n: number of patients. G: grade. RI: renal insufficiency. Thromb.: trombopenia. Neutr.: neutropenia.

## Data Availability

Data are available upon reasonable request.

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
