# Peer review of "Relevance of Therapeutic Drug Monitoring of Tyrosine Kinase Inhibitors in Routine Clinical Practice: A Pilot Study"

_pharmaceutics, 2022, doi:10.3390/pharmaceutics14061216_

Round 1

Reviewer 1 Report

In this paper, the authors evaluated the use of TDM for the guidance of using TKIs (erlotinib, imatinib, lapatinib, and sorafenib) in routine clinical practice for the treatment of cancer with anticipation to allow the use of TKIs in a more effective and safe way and avoid therapeutic failure and toxicity of these drugs in certain patients. However, I do not think the paper is prepared well and all the data are too simple and rough, which are not enough to support the conclusion. Therefore, the paper should not be published. Some major comments are as follows:

  1. The abstract is too long and very hard to read. The authors are suggested to shorten it to no more than 400 words, and the language needs to be more concise.
  2. The introduction is too verbose and illogical, the authors are recommended to rewrite it and highlight the key points of this paper. The part “1.1. Targeted therapies” is not needed to be written as the paper is not involved in targeted therapy. The introduction should not beyond 1000 words.
  3. In the results, there are too many tables, the authors need to prepare more visualized figures to make the results clearer and easier to understand.
  4. In figure 1, the font of figure text is too small and unclear. The authors are suggested to redraw the figure.
  5. Why only analyze the data of the first cycle and the last cycle? I think, the data from all the cycles are needed to be evaluated and analyzed.
  6. A control without TDM guidance should be analyzed and a significance analysis should be added.

Author Response

Thank you in advance for your observations.

  1. Done as suggested.
  2. Done as suggested. 
  3. We reviewed it, and tables proposed are considered neccessary in order to a better understanding of the results.
  4. We have changed the presentation of fig. 1
  5. We have analyzed the first and the last cycles for giving a better understanding of the magnitude of dose changes proposed, in line with the hypothesis of the work.
  6. It is a retrospective study, in the context of limited similar works, not a prospective one.

Reviewer 2 Report

I am glad to read your insightful article discussing TKI optimization recently used worldwide. It is efficient, and everyone wants to know how to maintain the TDM and their concentration! Here are some recommendations.

  1. Is there any reasonable explanation for why the concentration was increased at last in the sorafenib last cycle graph in Fig1, which was quite different from the others?
  2. The character size of rows 98 to 100 is smaller than the rest.
  3. Many similar expressions, such as "No significant..." can be described in other words.

Author Response

Thank you in advance for your observations.

  1. It´s a question of small "n", resulting in this paradoxical observation.
  2. Corrected.
  3. Done.

Reviewer 3 Report

Very interesting manuscript on TKI inhibitors in hematologic malignancies. Well written, in depth analysis of the literature, appropriate discussion. Only some minor check of the spelling and no more findings.

Author Response

Thank you for your observation. We hope that the present work can represent an impulse for TDM of TKIs and antineoplastic agents in general.

Reviewer 4 Report

The authors present the findings of a retrospective analytical study with clinical patient data to establish the importance of Therapeutic Drug Monitoring (TDM). Data from patients who received Tyrosine Kinase Inhibitor (TKI) treatment after TDM guidance and subsequent dosage adjustment recommendation have been compiled, analyzed and presented in the current work as a pilot study. The study is indeed interesting because of the context of work and its relevance with respect to routine clinical practice. The datas gathered for erlotinib, imatinib, lapatinib and sorafenib shows greater amount of variability among different individuals with respect to exposure and cycles subjected to monitoring. There are few datasets which are not conclusive largely based on the non-availability of references to enable comparisons and hence as authors indicated, the current study can be considered as one of the pilot studies trying to explore the feasibility and practical utility of TDM in clinical practice. Although there are not a lot of major concerns, the authors could have improved the data presentation format with respect to the organization of data, way of presentation and also include the adequate literature support. Few concerns in this regard and questions are given below.

1.         Please provide references to support statements indicated in lines 68-76 in the introductory section. This is important as it pertains to the significance of TDM and the criteria for including few classes of drugs as best candidates for TDM.

  1. “A high variability in the response to the TKIs is increasingly reported, due not only to the genetic heterogeneity of the pharmacological targets that determine tumor sensitivity, but also to the patient's pharmacogenetic background, such as cytochrome P450 polymorphisms and membrane transporters” – lines 104-105, in introductory section – provide valid literature support for these statements.
  2. Introductory section gives a very elaborate background on TDM relevance, factors of consideration, targeted therapies, variability issues etc. However, there are few sections which can be made shorter and more compact which would certainly improve the readability of the article – especially under ‘targeted therapies’. Authors could alternatively consider incorporating an illustration or a table to concise details and provide a gist of importance mechanism and factors of consideration.
  3. “The patients treated with lapatinib in this study had different tumors: breast cancer, GIST, pancreas, colon or esophagus, among others” – line 573, discussion section. Is there a specific reason due to which the authors did not consider a possibility of including patients from one tumor alone and for different TKIs so as to eliminate one variability factor relating to dose?
  4. Blood sampling times showed in table 1 is different for different drugs and authors indicate that it was based on optimal sampling theory and previous pharmacokinetic models. This needs better clarification. If they were based on previous studies, then the corresponding study for each drug needs to be cited separately.
  5. “ In 3 of the monitored cycles (8.6%) it was recommended to suspend the administration of the drug and to reevaluate it the clinical setting “ lines 370-371 for section 3.1.3 for lapatinib. ‘Revaluating it the clinical setting’ – what does it mean and how would it be revaluated? This needs better clarification.
  6. Please avoid reiterating statements relating to TKI mechanism in the discussion. The statements indicated in the introductory section need not be re-discussed. The manuscript needs to be more concise which would certainly improve the clarity.
  7. “For the other drugs included in this study, lapatinib and sorafenib, the % of patients who would most benefit from TDM were 52.1% and 43.2% respectively. There is no similar study in the literature to compare these results” – lines 699, 700 of discussion. These are important findings and hence would need further literature support. Please consider looking for literature exploring similar studies, PK studies with lapatinib and sorafenib if not TDM related studies?
  8. Please consider presenting Figure 1 in an alternate way of graphical representation. The very high variability is pretty evident from the graphs, but as it is an important dataset, authors could consider presenting it in a different form for better representation such as box plot, whisker plot or something similar.
  9. Authors have described the results in detail in the discussion section. However, they could try presenting it in a much more organized way under various subsections (may be for each drug in separate) for better readability and organization.
  10. What are the future prospects of this study or further considerations which can be taken into consideration for TDM studies with other TKIs from the light of findings from this study?
  11. Please follow consistency for abbreviation – HPLC: high performance liquid chromatography – lines 22 and 234.

Author Response

Thank you in advance for your observations.

  1. Done as suggested
  2. Done as suggested
  3. Done as suggested
  4. The main reason to do it as described is that are "real-wold" data. Different diagnoses and some times lapatinib administered with other drugs or pathological circumstances that can previsibly alter its metabolism.
  5. Done.
  6. Clarified in the new version as suggested.
  7. Done as suggested.
  8. We consider that such a comparison, not related with TDM studies, could be of limited value in the context analyzed.
  9. The presentation of fig. 1 has been corrected.
  10. Done as suggested.
  11. We currently extended the TDM to other TKIs (pazopanib, sunitinib) and explored the possibilities of obtaining financial support to acquire a mass spectometer.
  12. Done as suggested
  13.